# Cyber-Victimization Experience among Higher Education Students: Effects of Social Support, Loneliness, and Self-Efficacy

**DOI:** 10.3390/ijerph19127395

**Published:** 2022-06-16

**Authors:** Tali Heiman, Dorit Olenik-Shemesh

**Affiliations:** Department of Education and Psychology, The Open University of Israel, 1 University Road, Ranana 4353701, Israel; doritol@openu.ac.il

**Keywords:** cyber-victimization, higher education students, social support, loneliness, self-efficacy

## Abstract

Most of the research literature on cyberbullying (CB) has focused on adolescents, but due to their intensive, unsupervised use of Information Communication Technologies (ICT), higher education students are at high risk of being involved in CB. The current study examined the nature of CB among 1004 higher education students. In addition, we explored the relationships between cyber-victimization, social support, loneliness, and self-efficacy. For that purpose, we applied a path analysis model (PA) to explain the effect of each variable on the cyber-victimization experience, expecting that high levels of loneliness and low levels of self-efficacy will predict cyber-victimization, but might be moderated and reduced by high levels of social support. Results revealed that social support moderated the relationships between these socio-emotional variables and cyber-victimization, and might serve as a protective factor. These findings on young adults may contribute to the understanding of the nature of cyber-victimization throughout the life cycle. Nowadays, academic institutions are facing an uphill effort in trying to restrain online misbehavior. In view of the findings, higher education policy could help facilitate coping with CB through student support and focused intervention programs.

## 1. Introduction

Most of the research literature on cyberbullying (CB) has focused on adolescents, but due to their intensive, unsupervised use of Information Communication Technologies (ICT), higher education students are at high risk of being involved in CB. The current study examined the nature, patterns and characteristics of CB among higher education students. In addition, we explored the relationships between cyber-victimization, social support, loneliness, and self-efficacy. For that purpose, we applied a path analysis model (PA) to explain the effect of each variable on the cyber-victimization experience, expecting that high levels of loneliness and low levels of self-efficacy would predict cyber-victimization, but these might be moderated and reduced by high levels of social support.

### 1.1. Cyberbullying Victimization

Cyberbullying is a form of interpersonal aggressive behavior that occurs through electronic means, such as texting, online chats, or social media websites [1]. CB is also defined as deliberate, aggressive activity directed by an individual or a group aimed at hurting another person through digital means, such as the internet, mobile phones and other digital communication [2,3]. In contrast to other forms of bullying, it reaches a far wider audience at rapid speed, transcending boundaries of time as well as physical and personal space.

CB has a number of unique characteristics that often make it harder to cope with than face-to-face bullying: For example, *anonymity* allows an individual to share or disclose personal information about another without being exposed. Most cyber-victims do not know the identity of their attacker [4]; the *wide audience*—the Internet allows fast and broad distribution to a large number of users [4]. Those involved in CB can distribute messages, pictures or any other material to a wide audience [5]. *High accessibility* [6] and *difficulty of discovering and reporting*—cyber-victims are less likely to report bullying compared to victims of face-to-face bullying [4]. The high prevalence of CB among children and adolescents may be explained by the ‘disinhibition effect’ evident in online behaviors when, following the indirect nature of the act, as well as the anonymity, they are expressed using foul language, anger, threats and curses. Online behaviors lack the inhibition that occurs in the “real” world outside the internet [7]. The unique characteristics of the CB phenomenon make it hard to track and recognize online vulnerability, yet it has diverse negative socio-emotional and scholastic impacts on the victims [8,9,10,11].

Accompanying the widespread use of technologies, various studies have indicated increased incidence of CB among children and adolescents in various countries around the world [12,13,14,15]. Examining the changes in the use of digital technologies through the age developmental aspect, we can see similar trends across similar studies focusing on adolescents. For example, in data from the 2017 Youth Risk Behavior Survey in USA [14] among adolescents aged 14–18, 10% were victims of face-to-face bullying and CB. Similar findings were observed in a Canadian national study [16], wherein almost 14% of the children aged 10 to 17 had been cyberbullied, and within a cross-national study of seven European countries, for adolescents aged between 14–17, cyber-victimization ranged from 13.3% to 37.3% [16].

In Israel, a study we conducted on victims of CB among middle school students revealed that 16.5% of 242 students reported that they had been cyber-victims [11]. Two years later, in a large representative research study conducted during 2014 among Israeli children and adolescents [17], researchers found a significant increase in CB involvement, where 27% out of 1094 participants aged 9–17 reported being cyber-victims, 17% reported being cyber perpetrators and 46% reported they were bystanders in CB episodes.

Most studies in recent years have shown divergent negative socio-emotional and scholastic impacts on adolescents and children who reported being cyber-victims [10,11,18]. Studies revealed that mostly the social, emotional and behavioral areas are affected by cyberbullying such as frustration, apathy, loneliness, sadness, depression, anger, low self-esteem, difficulties in social adjustment or social withdrawal [18,19]. A meta-analysis conducted by Kowalski et al. [3] emphasizes a strong connection between CB and psychological variables. Moreover, most findings on the relationships between cyberbullying and gender have noted that more males tended to bully others compared to females and females tend to report on cyber-victimization more often than males [2,3].

### 1.2. Cyber-Victim Experience among Emerging Adults

For emerging adults (EA) or young adults (defined as the late teens and twenties), this period is when people obtain the level of education and training that provide the foundation for their career and income. EA often explore a variety of possible life directions in love, work, and worldview, thus experiencing years of profound change. EA are subjectively distinct, as they regard themselves as being neither adolescents nor adults [20]. For them, the criterion for adulthood is becoming self-sufficient. This is a period when they take on some of the responsibilities of independent living, but leave others to their parents, college or university authorities, or other adults [20]. EA tend to explore a wider scope of possible activities, as well as the highest rates of residential changes. Moreover, it is interesting to note that the degree to which participants in Lane’s study [21] identified with EA as being in a period of experimentation and possibilities was positively associated with well-being and life satisfaction, while EA identified the period between adolescence and adulthood as a time of instability, with negative feelings associated with decreased well-being and life satisfaction. Moreover, according to the study by Burger and Samuel [22], young people’s levels of stress and self-efficacy, as well as perceived stress and self-efficacy, are associated with life satisfaction, where baseline self-efficacy mitigated the negative effects of baseline stress on life satisfaction. Furthermore, identity formation mainly takes place for EA during this period, and identity explorations can be disappointing and result in rejection or failure [23]. It is worth mentioning that in the USA, participants aged 19–29 spend more time alone than any age group except for the elderly.

Focusing on young adults’ data from the Social Survey on Internet Usage during 2021 [24] shows that 90% of the Israeli population aged 20 and up use the Internet. In 2020, the most common uses of 90% of young people aged 20–44 were social networks, internet discussion, and searching for information. In Israel, Internet use among those aged 20 and over has increased over time. A decade ago, 68% used the Internet, compared to 90% in 2020.

In European studies, emerging adults who remain at home tend to be happier with their living situation than those who have left home [20]. Some studies demonstrated that high levels of parental involvement promote academic, athletic, and well-being outcomes, but detract from gaining independence. Support and meaningful connection can have a strong impact on family members’ well-being [25,26,27].

With the emergence and proliferation of new learning and social digital platforms in higher education via laptop or mobile smartphone, these opportunities also appear to increase CB experiences. Duggan and Smith (2013) reported that 73% of the adult students use at least one site online and about 42% of them use multiple social networking sites. The social networking sites are almost an inseparable part of the students’ lives, as they tend to share demographic and personal information. Studies show that the average college student shares over 100 photos on their Facebook gallery, while females posted three times more photos than males [28].

Studies conducted among college students revealed that over 90% of the Facebook users are more likely to disclose personal information on social networks [29]. Studies examining social network behavior/misbehavior among students in higher education institutions, indicated an increased involvement in CB, ranging from 10–35% [30]. Moreover, Paullet and Pinchot [31], revealed that 9% of the students were being cyberbullied, 57% of the victims were female and 43% were male. Another study among female college students [32] showed that 27.2% reported some involvement in CB as a bully, a victim, or both. The statistics of Varghese and Pistole’s [33] study reveal that 15% of university students reported CB victimization during college. Meanwhile, Giumetti and colleagues [34] found that 46.4% of the participants reported experiencing at least one act of cyber-victimization “at least monthly” or more often in the previous six months. For CB perpetration, 23.7% reported perpetrating one act of CB “at least monthly” or more often in the previous months.

It seems that the prevalence of CB varies immensely. For example, a study of international students in the USA showed that 20.7% of the international students had been cyberbullied in the previous 30 days once to many times: 45.5% reported that they had been cyberbullied in their lifetime once to many times [35]. In another study, using the New Zealand national report of a sample of 20,849 participants, findings revealed that participants in the 18–25 age group experienced the highest levels of CB [36]. In the UK [37] 46% of trainee doctors reported at least one instance of cyber-victimization, and among college students in Greece, it was found that 58.4% of the students were involved in CB [38]. Due to the small number of studies on CB among the EA population, the current study has examined its occurrence and vulnerability among students in this age group.

### 1.3. Predictors of Cyber-Victimization

The literature regarding personal, social, and emotional characteristics related to cyber-victimization highlighted various types of social difficulties, and loneliness as constituting potential risk factors that increase the probability of experiencing CB [11,39]. An examination of the relationships between CB and social-emotional aspects suggests that personal characteristics such as loneliness, low self-esteem and self-efficacy [40] and external characteristics such as low levels of social support [11] are viewed as risk factors related to cyber-victimization mostly among children and adolescents. Only a few studies have focused on higher education students regarding their involvement in CB and related to social and emotional aspects.

*Social support*. Perceived social support is defined as an individual’s perception of being cared for, valued and included by family, friends and peers, and may have an important role as a protective factor in the consequences of peer victimization [41]. Support sources such as a secure parental relationship and peer social support are positively associated with successful academic adjustment and social adaptation, and negatively associated with psychological symptoms of distress at the time of college entry [42]. Social support is often conceptualized as a protective factor in students’ lives that contributes to students’ successful adjustment to university, and has been found positively related to self-esteem [43], academic adjustment in college [44], and emotional well-being [45]. Family social support was identified as a significant positive factor for dealing with both CB and cyber-victimization, even for students who did not have supportive friendships [46]. Moreover, being cyberbullied was correlated to a low level of peer social support as well as low academic achievements [47]. The relationships between social support and CB were mainly examined among adolescents, where cyber-victimization was correlated to low social status, low self-esteem, and behavior difficulties in school and at home [12]. The risk of being a victim of CB has been associated with social and emotional difficulties, and loneliness [48]. Previous studies among adolescents with autism spectrum disorder, found that cyberbullying victimization was associated positively with peer rejection, anxiety, and depression [49] and similar patterns were found among adolescents with learning disabilities, while females are at higher risk of cyber-victimization, as well as having lower social support and were at higher risk of peer rejection compared to those without disabilities [11]. Akcil [34] suggest that it might be beneficial for researchers to explore variables such as social support, which may be helpful in developing counseling and university programs to support students dealing with mental health issues caused by CB. So far, only a few studies have focused on the effect of being involved in CB and social support among higher education students

*Sense of loneliness.* Many studies have highlighted the role of socio-emotional aspects in CB. Loneliness, peer rejection and lack of social support were found to be associated with involvement in CB [11,19]. The sense of loneliness has two categories: social and emotional. The former, described by the lack of companionship and the failure to establish stable social relationships, and the latter, referring to the experience of an unfulfilled need for close companionship, were found to be associated with involvement in online victimization. Cyberbullied adolescents had fewer friendships as well as lower ratings of social acceptance by peers. These results are similar to those of a study conducted within a sample of Israeli adolescents, in which findings revealed significant correlations between cyber-victimization, low sense of social and emotional loneliness, lower global self-worth and a more depressive mood compared to non-cyber-victims [8]. Significant correlations were found among Turkish cyber-victim students reporting on greater feelings of loneliness [48].

*Self-efficacy.* Self-efficacy is defined as one’s self-assessment of the degree of one’s ability to perform a particular task, a concept that relates to how an individual perceives him or herself, and believes in his or her own ability to organize and successfully perform certain actions in order to attain a desired result [50]. The concept of self-efficacy derives from socio-cognitive theory, according to which behavior, cognition and the environment affect each other dynamically; therefore, the concept of self-efficacy is dynamic and changes according to new information and new experiences acquired over time [51]. Self-efficacy is a factor affecting the individual’s decision-making process, goals, emotional reactions, effort, coping and persistence, and might change because of learning, experience and feedback [51]. There is considerable importance to the connection between self-efficacy and the involvement in cyberbullying. Self-efficacy was found to be strongly related to adolescents’ face-to-face bullying victimization and CB incidents, as the victims presented low perception of self-efficacy and believed they were unable to help other victims, or that others with greater self-efficacy might help the victims [52].

### 1.4. The Current Study

The goals in the current study are exploring the nature of CB in this sample, and to test the experiences of cyber-victimization in the context of personal and interpersonal aspects such as perceived self-efficacy, loneliness, and perceptions of social support, in order to identify the risk and protective factors among higher education students, as well as examining gender and disability differences. Within the current study, the relations between internet use, social-emotional variables, self-efficacy and cyber-victimization among higher education students is further examined. Based on the literature review, it was hypothesized that social support would affect social state (i.e., loneliness), or that the intensity level of online activity, regarding using various websites, would affect victimization probability via the mediation of social support and perceived self-efficacy. In addition, personal characteristics such as age, gender and disability were examined. It was hypothesized that women would report a higher level of experiencing CB than men, and that students with disabilities would report on more cyber-victimization compared to students without disabilities.

## 2. Methods

### 2.1. Procedure and Participants

After receiving approval from the ethics committee of the university (Ethic no. 2854), a Google Drive questionnaire was sent to 2000 students studying at the Open University of Israel (OUI), during the COVID-19 pandemic in Israel, through Spring semester April to June, 2021. The OUI institution functions nationwide across Israel, and is located in 60 study centers throughout Israel. The OUI students received a short email explaining the purpose of the study and guaranteeing the anonymity of the participant. All the participants agreed to participate in the study. Students who agreed to take a part in the study completed the questionnaires, which were entered into the data sheets. After three weeks, we had received 1013 responses. From this sample, we excluded nine students who did not complete specific background questions, or who reported they were not Hebrew speakers. The final sample consisted of 1004 students aged 18 to 35 (Mean = 27.15, SD = 4.03), 73.3% percent of the participants were students of social studies and humanities, 9.8% were students of life and exact sciences, and the remaining 16.9% were studying for a diploma. In addition, 67% were female and 33% were male. We also grouped respondents by their disability recognition, where 21% provided some sort of learning disability recognition. The distribution of the data obtained is representative by gender and corresponds to the OUI students’ direction of studying: 63% for social and humanities, 32% studying natural and exact sciences (OUI 2020 Report (https://www.openu.ac.il/en/about/president/presidentreport/pages/report2020.aspx (accessed on 1 April 2022).

### 2.2. Measures

The participants completed four questionnaires, and a short background inventory.

Student Survey of Cyberbullying [53]. The full questionnaire contains four parts: 1. Online activities—including nine items regarding using various websites, 2. The contribution of the internet to academic studies, 3. The students’ cyber-victimization experiences. Part 3 had 13 items regarding the experience of the victim during the last year, as: spreading offensive rumors; turning friends against victims; threats of physical harm; receiving sexual pictures or receiving sexual videos; receiving embarrassing posts; threats your family; insults; nasty rumors; tried to make your friends not love you anymore; mock you for what you looked like (e.g., from your body or your clothes); tried to hurt your dignity; someone sexually harassed you in a way that bothered you; someone presented himself as if he or she was you (to cause you harm). Part 4. Emotional experience of the cyber-victims. For this study, only parts 1 and 3 of the questionnaire were used. Both parts were answered on a 5-point Likert scale (0 = not at all, 4 = almost every day). Reliability for part 1 was α = 0.78 and for part 3 was α = 0.89.

*The construction of the cyber-victimization* binary *variable:* Due to highly skewed distribution of responses for cyber-victimization, we crosschecked these counts with a preliminary dichotomous question on whether or not the respondents experienced cyber-victimization. The final binary *cyber-victimization* was divided between no CB experience (*n* = 798, 79.5%) and CB experience (*n* = 206, 20.5%).

Multidimensional Scale Questionnaire for Perceived Social Support [54]. The questionnaire contained 12 items describing the participant’s current perception regarding the availability of social support from family, friends, or some other close, significant individual. Scale items were divided into three subscales and referred to support from (a) family, (b) friends, and (c) a close friend. For example, ‘my family tries to help me,’ ‘I can talk about my problems with my friend,’ and ‘I have a close person whom I can trust.’ Answers were given on a 7-point Likert scale (1 = ‘extremely unsuitable’, 7 = ‘extremely suitable’) with the high score indicating greater perceived social support. The reliability for the global support for the current study was α = 0.95.

Loneliness Questionnaire [55]. The questionnaire contained 16 items on a 5-point Likert scale ranging from ‘never feel like this’ (1) to ‘always feel like this’ (5). High scores reflect stronger feelings of loneliness. Sample items are, ‘I have many friends in my class’ and ‘I feel alone at school.’ The reliability of this sample was α = 0.92.

Self-efficacy [56]. The 13-item scale assessed students’ view of their academic and social self-efficacy. Answers were given on a 5-point Likert scale, ranging from 1 = ‘not at all’ to 5 = ‘very well’, with the high score reflecting a high sense of self-perception. Sample items are, ‘How well do you succeed in finishing all your academic assignments?’ and ‘To what degree are you able to connect with other students?’ The reliability for the global self-efficacy scale for the current study was α = 0.88.

The Background inventory included students’ characteristics such as age, gender, field of study and whether they were diagnosed with a disability.

### 2.3. Analytical Plans

For the research, we constructed four continuous indicators based on answers to relevant questions. The loneliness indicator was the mean of 17 items (mean = 2.76, SD = 0.72, reliability α = 0.92). The intensity level of online activity was based on ten online activities for which the respondents filled in the level of use from ‘none’ (0) to ‘daily’ (4) (mean = 1.59, SD = 0.58, reliability α = 0.78). We also calculated two other personal resilience measures based on the support from family and friends (mean = 5.90, SD = 1.32, reliability α = 0.95), and self-efficacy (mean = 3.62, SD = 0.67, reliability α = 0.88). Table 1 provides additional correlations between research variables.

## 3. Results

### 3.1. Preliminary Analyses

First, to examine the relationships between the students’ background characteristics of age, gender, and disability with the study variables of loneliness, online activity, social support, self-efficacy, and cyber-victimization (as a continuous variable), Pearson correlations analysis was conducted (see Table 1). The table shows correlations and descriptive statistics for students who did not experience cyberbullying and for those who did, separately.

Table 1 shows high correlations between loneliness and self-efficacy, which means that respondents, who reported greater loneliness, were more likely to report lower self-efficacy and vice versa. Similarly, support was negatively correlated with loneliness, whereas support was positively correlated with self-efficacy. In other words, external support from friends and family was positively correlated with self-efficacy and loneliness, and this can be seen as a resilience depowering.

Students’ disability was positively correlated with loneliness and negatively correlated with self-efficacy, that is, students with disability were lonelier and had lower self-efficacy. Note that although the latter correlations were found significant, correlation coefficients were considered low (r < 0.35).

The cyber-victimization variable was built as a binary response, based on the cyber experience respondents reported. Among students, who did not experience cyber-victimization, regarding gender differences, female students were significantly more online active in comparison to male students (t = −5.37, *p* < 0.001), and received high social support t = −3.93, *p* < 0.001.) Among those students who experienced cyber-victimization, the online activity was positively correlated with social support, though to a low degree. Student with learning disability were lower on self-efficacy in comparison to typical students. The comparative tests resulted in determined differences between the two groups. Specifically, as expected, loneliness and online activity were found higher among the cyber-victimization groups, on average, and this group was lower, on average, in social support and self-efficacy. No significant differences emerged for age and the study variables: loneliness, social support, self-efficacy and online activities.

### 3.2. Path Analysis Model

To further examine the potential effects of various research variables on the probability of experiencing cyber-victimization, we developed a path analysis model [57] in which we included a set of regression equations. Figure 1 shows the conceptual model, where the cyber-victimization is an endogenous variable. In this figure, in addition to direct effect, the dashed arrows represent indirect effect, or effects of an independent variable on the dependent variable via a mediator variable, specifically, the effect of social-emotional state (loneliness), or online activity (intensity level of online activity) on victimization probability via the mediation of social support and perceived self-efficacy. We expanded this indirect effect to include a moderated mediation, where self-efficacy was expected to moderate the mediation effect of social support.

The cyber-victimization outcome means a linear probability model rather than the common linear common approach. Specifically, the binary outcome is transformed into probability units, namely, the probability of experiencing cyber-victimization.

*Modeling results:* Figure 2 and Table 2 show modeling results, where Figure 2 shows only significant (*p* < 0.05) paths or model correlations, while Table 2 shows results for all direct paths [58,59].

In terms of personal characteristics, regarding gender, results show that women received more support and displayed higher online activity levels (β = 0.11, *p* < 0.001; β = 0.10, *p* < 0.01; respectively), while their self-efficacy was lower (β = −0.08, *p* < 0.01) than for the men. In terms of learning disability, students with disabilities were higher in loneliness and online activity (β = 0.08, *p* < 0.01; β = 0.07, *p* < 0.01; respectively). Those students with learning disabilities also had a greater probability of experiencing victimization by 60% or were 1.6 times more likely to experience cyber-victimization (OR = 1.60, *p* < 0.05) compared to students without any diagnosed disability. Similarly, exposure was associated more than twice to experience cyber-victimization over not, or 136% higher probability of experiencing victimization (OR = 2.36, *p* < 0.001). In other words, students who were more online active were more likely to experience victimization, and vice versa. In contrast, higher support was associated with lower probability of experiencing victimization by 31% (OR = 0.69, *p* < 0.001). In addition to these direct effects, an indirect effect from loneliness to victimization through support was found. Greater loneliness was associated with lower support, and lower support was associated with higher victimization (unstandardized indirect = 0.345; standardized indirect = 0.13, 95% CI [0.07, 0.18]; OR = 1.41; bootstrapping = 2000). Note, however, that this indirect effect represents linear association rather than probability. In probability terms, as suggested by Feingold, MacKinnon and Capaldi [57], we assessed the probability of victimization to be 41% higher than not being a victim in response to the indirect effect of loneliness through support.

We expanded the indirect effect test to include a moderation effect of self-efficacy, i.e., the interaction between support and self-efficacy. In other words, we looked at the indirect path: loneliness → support → victimization, moderated by the support*self-efficacy interaction effect on the probability of victimization. We expected this interaction to moderate the mediation effect, such that the latter would vary in response to varying values of self-efficacy or support. We found that the presence of the interaction in the model resulted in a slight change in the overall indirect effect (indirect = 0.387, 95% CI [0.19, 0.54]; OR = 1.47; bootstrapping = 2000). The interaction effect was found significant at *p* < 0.10 (β = −0.081, *p* = 0.093). However, the presence of varying values of self-efficacy did not affect the mediation level except when self-efficacy was high (unstandardized indirect = 0.31, *p* = 0.063). In contrast, when looking at varying values of support, we found that the indirect effect increased as support increased (Low support: unstandardized indirect = 0.20, *p* = 0.069; Moderate support unstandardized indirect = 0.39, *p* < 0.001; High support unstandardized indirect = 0.58, *p* = 0.001; bootstrapping = 2000).

Goodness-of-Fit indices based on the weighted least square mean and variance (WLSMV) estimator: χ2 = 0.68, df = 1, *p* = 0.41; CFI = 1.00, TLI = 1.00; RMSEA = 0.000, 95% CI [0.00, 0.08]; SRMR = 0.004.

## 4. Discussion

The aims of the present study were to examine the patterns and specific characteristics involved in cyber-victimization among young adults attending higher education institutions in Israel. Furthermore, we examined the contributions of social-emotional aspects, such as social support, loneliness, and perceived self-efficacy on experiencing cyber-victimization. As most of the research literature on social media and the Internet has focused on adolescents, the present study contributes to understanding the involvement of young adults in cyber-victimization. By conducting a path analysis model (PA), we can better explain the effect of each variable of the cyber-victimization on the young adult students.

As expected, findings demonstrated that social support moderated the relationships between the socio-emotional variables and cyber-victimization. In other words, reported high levels of loneliness and low levels of perceived self-efficacy might predict experiencing cyber-victimization, but high levels of social support might reduce and moderate the probability of being hurt by CB.

These results concur with previous studies examining the involvement of adolescents in CB episodes, and the effects of various social and emotional aspects. Most findings on children and adolescents showed that the risks involved in using online communication might be expressed through various aggressive behaviors against individuals or groups, including CB, texting improper content, shaming, and social exclusion [60,61]. Previous studies regarding experiencing cyber-victimization based on children’s and adolescents’ reports, e.g., [2,3,8] found that they described higher levels of emotional distress, loneliness, intensified feeling of frustration, helplessness, anxiety, and lower self-esteem compared to peers who were not cyber-victims. Similar patterns were found in the present study among young adults: the students who reported experiencing cyber-victimization also reported lower social support and lower self-efficacy, compared to peers who were not cyber-victims. As expected, social support moderated the effect of loneliness on cybervictimization. This result increased the importance of social support, which might be provided by the family members, peers or by close friends, not only during childhood and during adolescence, but also in in emerging adulthood.

Regarding gender differences, previous studies revealed that during adolescence, girls experience more cyber-victimization than boys, and boys were more perpetrators than girls [2,11], while only few studies found no gender differences for the interaction of online activities and victimization [40]. In the present study focusing on young adults, findings did show significant differences between genders. Woman reported spending more time on social networks, reported receiving higher social support compared to men, and perceived their self-efficacy lower than men. Social support is considered as a factor that moderates cyber-victimization, and a high social level of support might reduce risks of cyber-victimization. We may assume that social support might serve as a resilient and protective factor for women being online victims.

Regarding differences between students diagnosed with learning disabilities (LD), as shown previously [40,62], children and youth with mild disabilities were involved in CB as victims more often than students with typical achievements. The results of the present study on young adults are consistent with previous studies on Israeli adolescents [63]. In this study, young adults with LD attending higher education reported more Internet surfing hours and being lonelier compared to peers without LD. The SEM model results show a direct effect for the probability of experiencing cyber-victimization of young adults with LD. Therefore, they are at higher risk of being involved in CB and experiencing cybervictimization. The similarity of the findings regarding the different cycle of developmental age, childhood, adolescence and adulthood of individuals diagnosed with LD is one of the main contributions of the present study.

## 5. Conclusions

As only sparse studies have been conducted regarding the online experiences of students with LD attending higher education, their involvement and vulnerability as related to social-emotional feelings of loneliness and being at higher risk for social and emotional experience might be one of the barriers faced by individuals with LD to being successfully involved in the social environment. Yet, the present study did not differentiate between social loneliness and emotional loneliness. Further study is needed to conduct a deeper examination of students diagnosed with LD. In addition, further studies should examine the severity of students with LD, as they might have a higher predisposition to be more vulnerable or at higher risk of being cyber-victims than other students.

These findings about young adults may contribute to the understanding of the nature of cyber-victimization throughout the life cycle. Nowadays, schools and higher education institutions are facing an uphill battle trying to restrain online misbehavior. In view of the findings, developing intervention programs for different age groups could help facilitate coping with CB, as well as implementing effective tools for coping with social networks and to enhance social support through all age periods.

## Figures and Tables

**Figure 1 ijerph-19-07395-f001:**
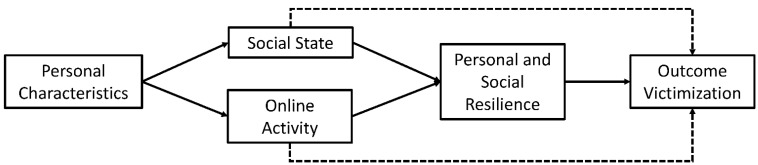
The theoretical structural model, direct and indirect effects. Note: Straight arrows for direct regression effects; dashed arrows for indirect regression effects.

**Figure 2 ijerph-19-07395-f002:**
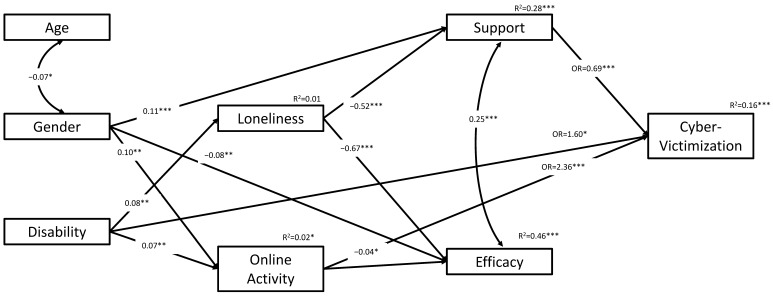
Path Analysis results for the probability of experiencing cyber-victimization. **Note.** *** *p* < 0.001, ** *p* < 0.01, * *p* < 0.05; Straight one head arrow for regression path, double-arrow arch for correlation.

**Table 1 ijerph-19-07395-t001:** Correlations between research variables and descriptive statistics.

	Disability ^1^	Loneliness	Online Activity	Social Support	Self-Efficacy	Cyber-Victimization
Age	−0.02	−0.04	−0.01	−0.001	−0.02	
Gender	−0.03	−0.07	0.12 **	0.18 ***	−0.04	
Disability	-	0.02	0.06	0.01	−0.02	
Loneliness	0.20 **	-	−0.07	−0.49 ***	−0.65 ***	
Online Activity	0.04	−0.10	-	0.06	−0.02	
Social Support	−0.05	−0.55 ***	0.16 *	-	0.49 ***	
Self-Efficacy	−0.22 **	−0.71 ***	0.06	0.45 ***	-	
Means	0.19	2.70	1.53	6.06	3.68	No: 798, 79.5%
SD	0.39	0.69	0.57	1.21	0.65	
Means	0.29	2.96	1.81	5.32	3.41	Yes: 206, 20.5%
SD	0.46	0.79	0.56	1.56	0.69	
	χ^2^(1) = 9.99 ***	t = −4.17 ***	t = −6.30 ***	t = 6.29 ***	t = 5.25 ***	

*** *p* < 0.001, ** *p* < 0.01, * *p* < 0.05. Note. The upper triangular shows correlations for students who did not experience cyber-victimization, and the lower triangular for those who did. Means and SDs are divided similarly followed by a comparative test (*t*-test for the continuos measures, and Chi-square test for disability proportion). ^1^ Disability-high score are students with disabilities.

**Table 2 ijerph-19-07395-t002:** Modeling results, standardized regression coefficients and correlation.

	Loneliness	Online Activity	Social Support	Self-Efficacy	Cyber-Victimization
	Sca5	Sca1	Sca4	Sca6	
Age	−0.03 (0.03) 95% CI [−0.08, 0.05]	−0.03 (0.03) 95% CI [−0.11, 0.03]	0.01 (0.03) 95% CI [−0.04, 0.06]	−0.03 (0.03) 95% CI [−0.08, 0.02]	0.97 (0.02) 95% CI [0.94, 1.01]
Gender	−0.05 (0.03) 95% CI [−0.13, 0.01]	0.10 ** (0.03) 95% CI [0.05, 0.17]	0.11 *** (0.02) 95% CI [0.07, 0.16]	−0.08 ** (0.02) 95% CI [−0.12, −0.03]	1.30 (0.22)9 5% CI [0.99, 1.92]
Disability	0.08 ** (0.03) 95% CI [−0.01, 0.13]	0.07 * (0.03) 95% CI [0.03, 0.13]	0.02 (0.03) 95% CI [−0.03, 0.06]	−0.04 (0.03) 95% CI [−0.08, −0.01]	1.60 * (0.29) 95% CI [1.26, 2.54]
Loneliness		−0.04 (0.03) 95% CI [−0.10, 0.02]	−0.52 *** (0.03) 95% CI [−0.57, −0.47]	−0.67 *** (0.02) 95% CI [−0.70, −0.62]	0.98 (0.17) 95% CI [0.73, 1.33]
Online Activity			0.000 (0.03) 95% CI [−0.05, 0.05]	−0.04 * (0.02) 95% CI [−0.08, −0.01]	2.36 *** (0.39) 95% CI [1.70, 3.39]
Social Support				0.25 *** (0.03) 95% CI [0.20, 0.32]	0.69 *** (0.06) 95% CI [0.60, 0.80]
Self-Efficacy					0.81 (0.14) 95% CI [0.59, 1.19]
R^2^	0.01 (0.01)	0.02 (0.01)	0.28 *** (0.03)	0.46 *** (0.03)	0.16 *** (0.03)

*** *p* < 0.001, ** *p* < 0.01, * *p* < 0.05; Standard errors in parentheses; Odd Ratio for cyber-victimization; 95% CI for 95% confidence interval (in squared brackets) around the point estimate; Shaded cells for correlations.

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
