# Peer review of "Cyber-Victimization Experience among Higher Education Students: Effects of Social Support, Loneliness, and Self-Efficacy"

_ijerph, 2022, doi:10.3390/ijerph19127395_

Round 1
Reviewer 1 Report
Dear Authors,
I appreciated your efforts to improve the quality of the manuscript.
Nevertheless, I still have many concerns about statistics:
1) it remains quite 'strange' the use of the Pearson correlation coefficient (but then, in table 1, you refer to Spearman) for evaluating the linear relationship between continuous variables and gender; i.e it is hard for me to interpret a correlation coefficient of -0.12 between age and gender when it would be so simple to use a t-test (or a non-parametric test) to assess differences in mean (or median) age between males and females. Moreover, when one of the variables is dichotomous a Point-Biserial Correlation should be used.
What about comparing males and females with the appropriate statistical tests for the study variables and reporting only the most relevant results (i.e. if and when you find differences statistically significant)?
2) It also sounds quite strange you feel the need to explain the meaning of odds and OR. Again, I find the sentences in the paper quite confusing: p/(1-p) is the odd formula; the OR compares that 'probability' between 2 groups.
In this perspective, it's incorrect the phrase, that seems to be referred to OR "...this ratio gives the probability of experiencing victimization over the probability of not experiencing" as, again, this is the definition of the odd.
Author Response
Journal IJERPH (ISSN 1660-4601)
Manuscript ID ijerph-1767903
Type
Article
Title
Cyber-victimization experience among higher education students: Effects of social support, loneliness, and self-efficacy
1) it remains quite 'strange' the use of the Pearson correlation coefficient (but then, in table 1, you refer to Spearman) for evaluating the linear relationship between continuous variables and gender; … when it would be so simple to use a t-test to assess differences ...What about comparing males and females with the appropriate statistical tests for the study variables and reporting only the most relevant results (i.e. when you find differences statistically significant)?
Response: We deleted gender and age from Table 1, and instead, as suggested, we use a t-test to assess gender differences and we reported only the most relevant results. No significant differences emerge for age.
2) It also sounds quite strange you feel the need to explain the meaning of odds and OR. Again, I find the sentences in the paper quite confusing: p/(1-p) is the odd formula; the OR compares that 'probability' between 2 groups.
In this perspective, it's incorrect the phrase, that seems to be referred to OR "...this ratio gives the probability of experiencing victimization over the probability of not experiencing" as, again, this is the definition of the odd.
Response: We deleted the phrase on page 8.
Thank you for your comments.

Reviewer 2 Report
I am grateful to the authors that they supplemented the article with the necessary data. I believe the article in its current form can be recommended for publication.
P.S. Despite the fact that loneliness is more often attributed to negative experiences, there is still experience in researching its positive aspects:
Long C.R., Averill J.R. Solitude: An exploration of benefits of being alone // Journal for the Theory of Social Behavior. 2003. 33. 1. 21–44.
Storr A. Solitude: A return to the self. N.Y.: Free Press, 2005.
Houghton S, Hattie J, Carroll A, Wood LJ, Baffour B. It Hurts To Be Lonely! Loneliness and Positive Mental Wellbeing in Australian Rural and Urban Adolescents. Journal of psychologists and counsellors in schools. 2016; 26(1): 52-67. DOI:10.1017/jgc.2016.1.
Author Response
Response to Reviewer
1) it remains quite 'strange' the use of the Pearson correlation coefficient (but then, in table 1, you refer to Spearman) for evaluating the linear relationship between continuous variables and gender; … when it would be so simple to use a t-test to assess differences ...What about comparing males and females with the appropriate statistical tests for the study variables and reporting only the most relevant results (i.e. when you find differences statistically significant)?
Response: We deleted gender and age from Table 1, and instead, as suggested, we use a t-test to assess gender differences and we reported only the most relevant results. No significant differences emerge for age.
2) It also sounds quite strange you feel the need to explain the meaning of odds and OR. Again, I find the sentences in the paper quite confusing: p/(1-p) is the odd formula; the OR compares that 'probability' between 2 groups.
In this perspective, it's incorrect the phrase, that seems to be referred to OR "...this ratio gives the probability of experiencing victimization over the probability of not experiencing" as, again, this is the definition of the odd.
Response: We deleted the phrase on page 8.
Thank you for your comments.
Please see the attachment file of the corrected paper, marked in YELLOW.

This manuscript is a resubmission of an earlier submission. The following is a list of the peer review reports and author responses from that submission.
Round 1
Reviewer 1 Report
I thank the Editors for the opportunity to review this manuscript and the authors for their hard work. However, I have many concerns about the paper in its current form and especially about the analyses and presentation of the results.
- First of all, I have found here and there phrases in the text (in the Introductiont, at the end of section 1.2, at the beginning of chapter 1.3 and finally in paragraph 1.4) that refer to the objectives of the study that should be collected in a single paragraph, so that aims are clearly pointed out.
- In 2.1. Participants section it is not clear where the final sample comes from; it is specified in the following 2.2 Procedure, but it should be clear from the outset who the sample is made up of, not just its size. I would suggest merging the current two paragraphs mentioned above into one.
- Page 2: references for "various studies conducted in recent years..." are only two and they refer to studies published in 2009 and 2010 which do not appear to be various or recent
- Pearson correlations in table 1: it is not clear how it is possible to calculare a Pearson correlation coefficient with qualitative variables such as gender; moreover, it is not clear to me if the variable Cyber-victimization variable is here continuous or already dichotomized. In this second case there is the same problem as for gender. In statistics, 'correlation' indicates an association between two quantitative variables and it assumes that this association is linear (i.e. age and blood pressure, heights and weights,...). Based on this definition, it cannot be used with qualitative variables such as gender (or cyber-victimization if dichotomized).
- In Modeling results section authors refer to odds and ORs and define OR as p/(1-p) which is actually the definition of odd... Moreover, using sometimes odds and sometimes ORs in the text is confusing for the reader, as well as confusing it is to interpret an OR=2.36 as a 136% higher probability of...
- In my opinion Table 2 is very difficult to read and even more difficult to understand. Again, the authors refer to correlation, but those for cyber-victimization should be ORs. And ORs cannot be negative! (see -.69 for social support). I strongly suggest thinking another way to present these results.
Minor revisions are due for spelling
Hope the authors can answer my concerns.
Reviewer 2 Report
The problem of cyberbullying among students raised by the authors of the article is among the most relevant in the modern world. Unfortunately, it is difficult to cope with it both from a legal and psychological point of view. The authors conducted a thorough theoretical analysis and formulated hypotheses that are tested in the article. The results are presented and discussed. Conclusions are drawn.
However, there are several issues that need to be clarified. Firstly, the article does not present data on the distribution of the data obtained: do they correspond to normal in the male and female parts of the sample? There is no data on the representativeness of the sample (for example, by the direction of training: natural or humanitarian directions). The presented data also need clarification: what are the consistency parameters of the SEM model? All data must be submitted. The article does not present comparative data on students who have experience of cyber-victimization and do not have it, although this is discussed in the discussion. As a debatable question: do the authors believe that loneliness has an exclusively negative side? Or was the questionnaire a purely technical fixation of the lack of social ties?
Round 2
Reviewer 1 Report
Dear Authors,
I appreciate your efforts to improve the quality of your manuscript and to answer my suggestions and concerns. However, I don't feel that the analysis plan is still enough strong for the paper to be published:
- I can still see correlation coefficients between gender and other continuous variables, that it doesn't make sense
- I think you are confusing ODD and OR as a measure of association: i.e. the measure of association for the two variables "students with learning disabilities" and "experiencing victimization" should be an OR. In fact, you write " 1.6 times more likely to experience..." that is an OR
- I am perfectly aware of the meaning of an OR=2.36 (136% higher risk); nevertheless, in my previous comments I was wondering if this explanation (that uses %) could be enough clear for the readers, clearer than "a risk 2.4 times higher"
Author Response
24 April 2022
Re: Cyber-victimization experience among higher education students: Effects of social support, loneliness, and self-efficacy
We would like to thank the reviewer for the comments.
Please find our attached file and our corrections based on the reviewer's recommendations.
